# Mmu-let-7a-5p inhibits macrophage apoptosis by targeting CASP3 to increase bacterial load and facilities mycobacterium survival

Xuehua Zhan[1], Wenqi Yuan[1], Rong Ma[1], Yueyong Zhou[2], Guangxian Xu[3], Zhaohui Ge[1]*

1 Department of Orthopedics, General Hospital of Ningxia Medical University, Yinchuan, Ningxia, China,
2 Clinical Medicine School, Ningxia Medical University, Yinchuan, Ningxia, China, 3 The First Dongguan Affiliated Hospital, Guangdong Provincial Key Laboratory of Medical Molecular Diagnostics, School of Medical Technology, Guangdong Medical University, Dongguan, China

* myovid@126.com

## Abstract

We have been trying to find a miRNA that can specifically regulate the function of mycobacterial host cells to achieve the purpose of eliminating Mycobacterium tuberculosis. The purpose of this study is to investigate the regulation of mmu-let-7a-5p on macrophages apoptosis and its effect on intracellular BCG clearance. After a series of in vitro experiments, we found that mmu-let-7a-5p could negatively regulate the apoptosis of macrophages by targeting Caspase-3. The extrinsic apoptosis signal axis TNFR1/FADD/Caspase-8/Caspase-3 was inhibited after BCG infection. Up-regulated the expression level of mmu-let-7a-5p increase the cell proliferation viability and inhibit apoptosis rate of macrophages, but down-regulated its level could apparently reduce the bacterial load of intracellular Mycobacteria and accelerate the clearance of residual Mycobacteria effectively. Mmu-let-7a-5p has great potential to be utilized as an optimal candidate exosomal loaded miRNA for anti-tuberculosis immunotherapy in our subsequent research.

**Data Availability Statement:** All relevant data are within the manuscript and its Supporting Information files.

## 1. Introduction

The incidence of spinal tuberculosis accounts for 50% -75% of Bone and joint tuberculosis (BJTB) [1, 2]. Traditional anti-tuberculosis drug therapy combined with surgical intervention is commonly utilized for the treatment of spinal tuberculosis. Due to the structure and pharmacokinetic characteristics of first-line anti-tuberculosis drugs, conventional dosage forms are difficult to penetrate the bone marrow-blood barrier (MBB), which resulting in poor anti-tuberculosis efficacy [3, 4]. How to efficiently deliver anti-tuberculosis drugs to the core area of tuberculosis foci has always been a research difficulty in this field. Exosomes are a class of extracellular vesicles which could contain numerous items derived from parental cells and shuttle them to receipt cell [5]. Exosomes can not only be used as capable transport vehicles to

**Funding:** This work was supported by the National Natural Science Foundation of China [82360434 to Zhaohui Ge] and the School Level Project of Ningxia Medical University [XT2023042 to Xuehua Zhan]. The funders had no role in study design, data collection and analysis, decision to publish, or preparation of the manuscript.

**Competing interests:** The authors have declared that no competing interests exist.

**Abbreviations:** CFU, Colony forming unit; EDTA, Ethylenediaminetetraacetic acid; ELISA, Enzyme-linked immunosorbent assay; FADD, Fas associated via death domain; FITC, Fluorescein isothiocyanate; IL-6, Interleukin 6; IL-1β, Interleukin 1 beta; IL-10, Interleukin 10; OD, Optical density; PBS, Phosphate buffered saline; TB, Tuberculosis; TBST, Tris-Buffered saline tween-20; TEM, Transmission electron microscopy; TNFR1, Tumor necrosis factor receptor superfamily member 1A; UC, Ultracentrifugation.

deliver traditional drugs, but also the genetic drugs with immunomodulatory functions such as miRNAs and proteins [6–10].Currently, research based on exosomes or extracellular vesicle therapy mainly focused on immune regulation, nervous system diseases, tissue repair and regenerative medicine [11–14]. However, there is no related report on the treatment of spinal tuberculosis.

The clearance of Mycobacterium tuberculosis in lesions is the core process throughout the whole course. It has been shown that the apoptosis of macrophages is closely related to the prognosis and outcome of mycobacteria. Apoptosis of macrophages in the early stage of infection could inhibit the increase of intracellular bacterial load and facilitate the clearance of Mycobacteria [15–17]. In our previous study, we found that mmu-let-7a-5p and its targeted gene Caspase-3 were associated with the regulation of cell growth and death, especially in apoptosis. Therefore, we proposed the bold idea of constructing a novel exosomal drug delivery system loading with rifampin and mmu-let-7a-5p for anti-tuberculosis therapy. It is hoped that traditional anti-tuberculosis drugs can be delivered pass through various biological impediments to increase the drug concentration in spinal tuberculosis areas by the aid of the nano size and low immunogenicity of exosomes. At the same time, the loaded miRNA molecules could also play its anti-tuberculosis immune regulation to accelerate the clearance of mycobacteria by macrophages in order to achieve the multiple purposes of anti-tuberculosis infection. Whereupon, the function of macrophages apoptosis regulated by mmu-let-7a-5p is crucial for our subsequent research.

In the present study, we investigated the influence of mmu-let-7a-5p on macrophages apoptosis and further validated its effect on the clearance of intracellular mycobacteria load in vitro.

## 2. Methods

### 2.1 Cell & mycobacteria culture and infection assay

RAW264.7 cells and Mycobacterium bovis Bacillus Calmette-Guérin (BCG) St. Pasteur 1173P2 strain were cultured as we described in our previous study [18], and the infection assay was followed with Bettencourt's description [19, 20].

### 2.2 Bioinformatics analysis

Metascape (https://metascape.org) and CyotoHubba in Cytoscape (v3.1.2) were selected to complete the visual analysis of the target genes related to cell growth and death. TargetScan (Mouse v8.0) and DIANA TOOLS were used to predict the seed sequence of mmu-let-7a-5p targeted binding with the 3′UTR of Caspase-3.

### 2.3 Synthesis of mimics/inhibitor and siRNA & construction of vector

The synthesis of mmu-let-7a-5p mimics, inhibitor and Caspase-3 siRNA and the construction of Caspase-3 expression plasmid were commissioned by Shanghai Gene-Optimal Biological Technology Co., Ltd. The transfection efficiency was determined by fluorescence detection of the transfected cells after construction.

### 2.4 Cell transfection

A total of 300 uL Opti-MEM I Reduced Serum Medium was transferred into 1.5 mL EP tube, 3ug plasmid or synthetic RNA was added or co-transfected into EP tube and mixed well. According to the instructions, a volume of 9uL Fu GENE ® HD Transfection Reagent was

applied into EP tube continuously. The mixture was incubated at room temperature for 15 min and added into the wells subsequently. Fluorescence was observed after 24 hours.

## 2.5 Dual luciferase reporter gene assay

The 3'UTR- wild type (WT) / mutant type (MUT) sequence of Caspase-3 was constructed into the pmirGLO reporter vector. The mixture containing the reporter vector and mmu-let-7a-5p mimics or inhibitor was co-transfected into different groups. The dual luciferase Reporter Gene assay kit (Beyotime, Shanghai, China) was chosen for detection according to the instructions.

## 2.6 Cell Counting kit-8

Cells were grouped and transfected with mmu-let-7a-5p mimics / inhibitor and Caspase-3 siRNA / OE followed by BCG stimulation for 4 hours, then washed with PBS. Set infection starting point as 0h, Cell Counting Kit (Beyotime, Shanghai, China) was selected to detect the OD values of cells at different time points post infection.

## 2.7 Flow cytometry

Cell grouping and transfection process were the same as the CCK-8 experiment. Annexin V-FITC/PI (Solarbio, Beijing, China) was used to examine the apoptosis rate. Briefly, cell suspension was collected after digestion with trypsin without EDTA, centrifuged at 1000 rpm for 5 min, added with 1 mL pre-cooled PBS, centrifuged at 1000 rpm for 5 min, and cell precipitation was retained. After resuspending, the cell concentration was adjusted to $5 \times 10^6$ / mL. 100 μL cell suspension was transferred into 5 mL tube, 5 μL Annexin V / FITC was added to mix for 5 min. 5 μL PI solution and 400 μL PBS were added for flow cytometry immediately.

## 2.8 qRT-PCR

The RNAeasy™ Animal RNA Isolation Kit with Spin Column (Beyotime, Shanghai, China) was used to extract RNA, and the SanPrep Column microRNA Extraction Kit (Sangon Biotech, China) was used to extract miRNA accordance with the instructions. RNA reverse transcription, cDNA verification, preparation of Real time-PCR reaction system and the setting of reaction conditions were the same as previous [18]. GAPDH was a reference gene for qPCR detection of mRNA. The primers information of related genes is listed in Table 1.

## 2.9 Western blotting

Membrane and Cytosol Protein Extraction Kit (Beyotime, Shanghai, China) and Enhanced BCA Protein Assay Kit (Beyotime, Shanghai, China) were used to complete the extraction and concentration determination of total proteins. The PVDF membrane was activated by methanol before SDS-PAGE electrophoresis.

The filter paper, gel and PVDF membrane were prepared correctly and then 300 mA constant transfer was started. TNFR1 Rabbit Polyclonal Antibody(1:1000 dilution, AF8196, Beyotime, Shanghai, China); Caspase-3 antibody (Rabbit polyclonal antibody) (1:1000 dilution, AC030, Beyotime, Shanghai, China); Caspase-3 (activated) antibody (Rabbit mAb) (1:1000 dilution, AC033, Beyotime, Shanghai, China); and GAPDH Rabbit Monoclonal Antibody (1:1000 dilution, AF1186, Beyotime, Shanghai, China)were incubated at 4˚C overnight. Membranes were incubated with HRP-labeled Goat Anti-Rabbit IgG (1:2000 dilution, Proteintech, USA) as secondary antibody after three times washing. Finally, chemiluminescence detection was completed after primary and secondary antibody incubation.

**Table 1. List of miRNAs and mRNAs primers information.**

| Primer name | Primer sequence |
|---|---|
| mmu-let-7a-5p | RT: CTCAACTGGTGTCGTGGAGTCGGCAATTCAGTTGAGAACTATAC |
| | F: ACACTCCAGCTGGGTGAGGTAGTAGGTTGT |
| mmu-U6 | RT: CGAATTTGCGTGTCATCCTTG<br>F: CTCGCTTCGGCAGCACATATAC |
| Reverse (universal) | TGGTGTCGTGGAGTCG |
| Caspase-8 | F: GCAGAAAGCGAAGCAGCCTA<br>R: AGGTTTGCTACCGATTCCGA |
| FADD | F: GTTCCTTGGGGGAAGACACC<br>R: GGCCTCCAAGGATGTGAGAC |
| TNFR1 | F: AAGGCTGGAAAGCCCCTAAC<br>R: AGAACTAAAGCCTGGGGTGC |
| GAPDH | F: AGGTTGTCTCCTGCGACTTCA<br>R: TGGTCCAGGGTTTCTTACTCC |
| Caspase-3 | F: CAGCCAACCTCAGAGAGACA |
| | R: ACAGGCCCATTTGTCCCATA |

## 2.10 ELISA

The expression of TNF-α, IL-6, IL-1β and IL-10 in each group at 48 h after BCG infection were detected by different Enzyme-linked immunosorbent assay (ELISA) Kit (Jianglai Biotech, Shanghai, China). The sample preparation, washing and antibody incubation steps were followed strictly by the instructions. The OD value was detected and the real concentration of the sample was calculated.

## 2.11 Colony forming unit determination

The infection process was the same as previous[18, 20]. The DMEM containing antibiotics was replaced for further culture for 48 h. The lysis cells were serially diluted and coated with Mycobacterium Solid Culture Medium (Gene-Optimal, Shanghai, China). The cells were further cultured at 37˚C for 2–3 weeks, and the number of colonies in the culture dish was counted.

## 2.12 Statistical analysis

All experiments were set at least three biological repetitions. The data were displayed as mean ± standard deviation. Graphpad Prism 9.0 (GraphPad Software Inc., CA, United States) was used for statistical analysis. Non-paired t test was adopted between the two groups and one-way analysis of variance (ANOVA) in the multiple groups to compare the differences. $p < 0.05$ was considered to be statistically significant.

# 3. Results

## 3.1 The visual analysis and Hub genes

The Metascape visual analysis results indicated that these genes were involved in multiple biological processes and pathways and correlated with the regulation of apoptosis (Fig 1A). Caspase-3, Bcl2l1, Casp8, Xiap, Akt1, Apaf1, Birc3, Bcl2l11, Mcl1 and Diablo were the Hub genes obtained by using CytoHubba for the target genes involved in apoptosis (Fig 1B).

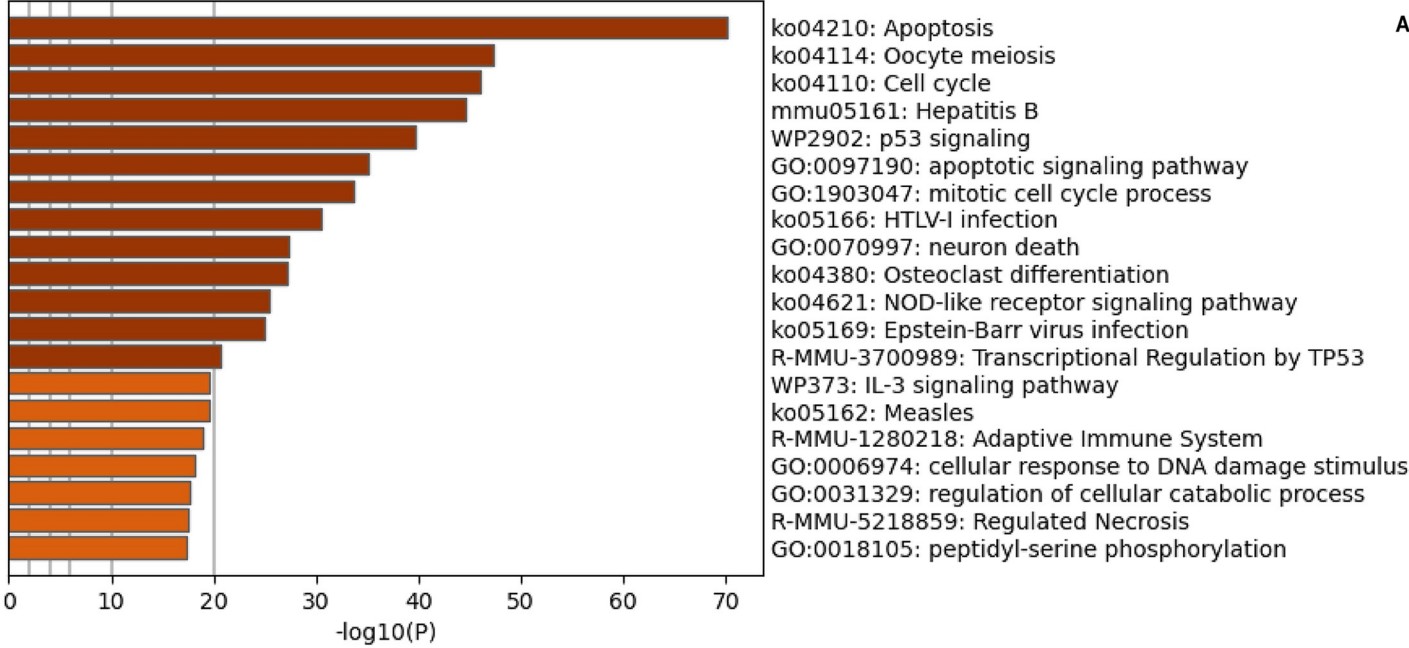

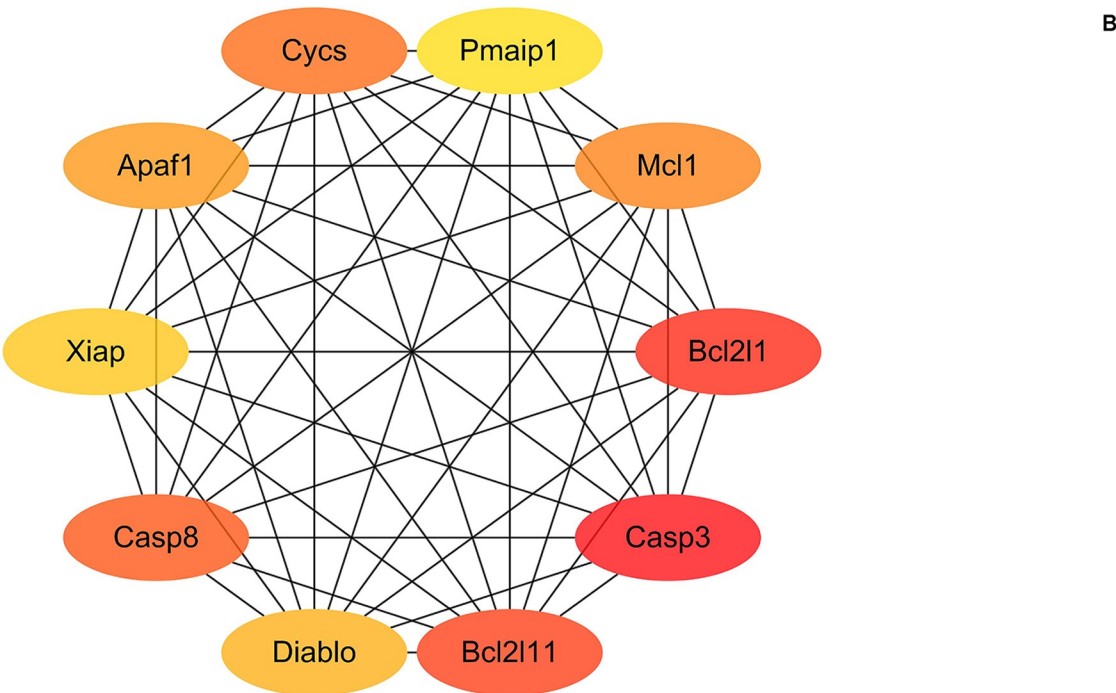

**Fig 1. The visual analysis and Hub genes.** (A) The visual enrichment of top 20 terms of target genes related to cell growth and death by Metascapse. (B) Hub genes participated in apoptosis filtered by Cytohubba in Cytoscape (v3.2). The darker the colour of the gene, the higher weight of the gene in the apoptosis process.

## 3.2 Mmu-let-7a-5p targets 3′UTR of Caspase-3

Bioinformatics analysis predicted that Caspase-3 might be the target gene of mmu-let-7a-5p and the seed sequence of mmu-let-7a-5p was completely complementary to the 152–159 base of the 3′UTR of Caspase-3 in TargetScan (v8.0) (Fig 2A).

## 3.3 Verification of mmu-let-7a-5p mimics / inhibitor & Caspase-3 siRNA / OE

After the synthesis of mmu-let-7a-5p mimics / inhibitor gene was completed, the transfection efficiency of Raw264.7 cells was detected to be over 80%. Raw264.7 cells were transfected with NC-mimics, let-7a-5p-mimics, NC-inhibitor and let-7a-5p-inhibitor, and then the expression level of mmu-let-7a-5p was detected by qRT-PCR. U6 was used as an internal reference. As shown in Fig 3A, that the expression level of intracellular mmu-let-7a-5p in Raw264.7 cells was significantly changed after transfection of mimics and inhibitor, respectively. Compared with the NC-inhibitor group, the expression level of let-7a-5p in let-7a-5p inhibitor group was significantly down-regulated, and the difference between groups was statistically significant ($p < 0.05$). The expression level of let-7a-5p in let-7a-5p mimics group was significantly up-regulated compared with the NC-mimics group. The difference between groups was statistically significant ($p < 0.001$). The consequences certificated that mmu-let-7a-5p mimics and inhibitor gene synthesis sequence was correct, cell group transfection was successful and reached the expected requirements of the experiment, which can be used for subsequent experiments. After the construction of Caspase-3 over-expression plasmid (Casp3-OE) and the synthesis of siRNA (Casp3-siRNA), the mRNA and protein levels of Caspase-3 were also detected

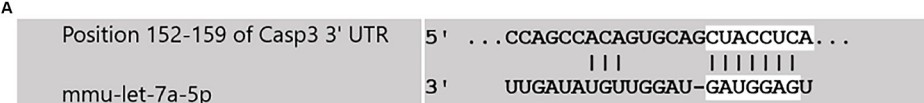

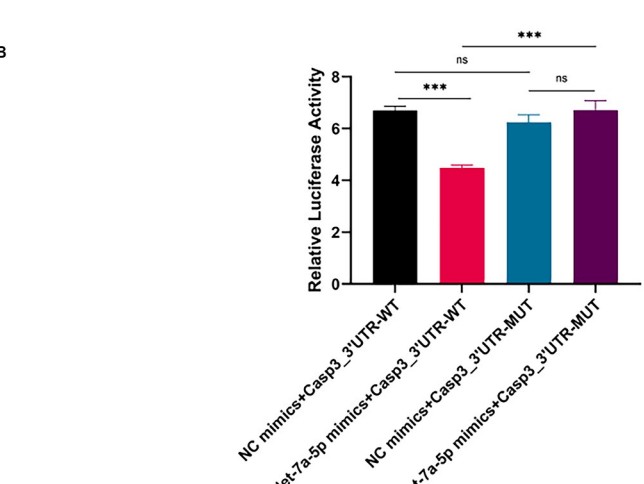

**Fig 2. The results of the dual luciferase gene reporter assay.** (A) Predicted consequential pairing of target region (top) and miRNA (bottom) in TargetScan(v8.0). (B) The relative luciferase activity normalized to Renilla luciferase activity was detected by the dual luciferase gene reporter assay. *** $p < 0.001$. n = 3 repeats.

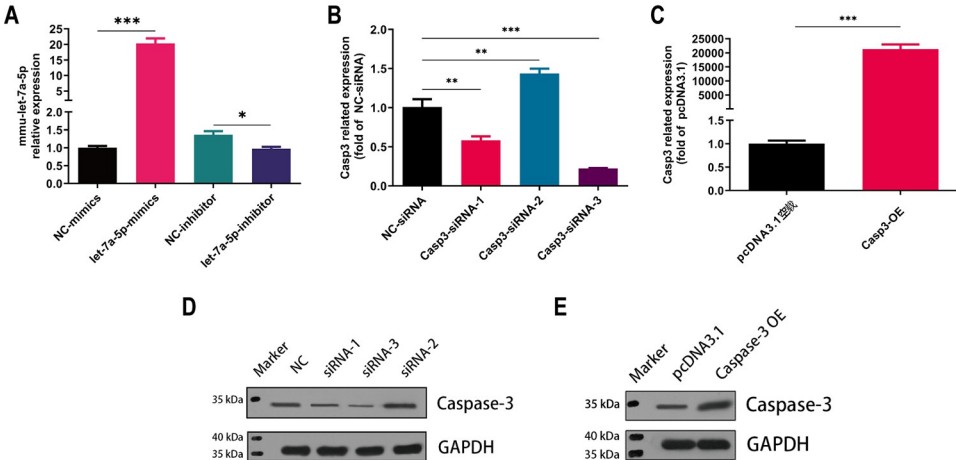

**Fig 3. Validation of synthetic chemicals and construction vectors.** (A) qPCR verification of mmu-let-7a-5p mimics and inhibitor, (B) qPCR verification of three Caspase-3 siRNA, (C) qPCR verification of Caspase-3 Over expression plasmid (Casp-3 OE), (D) Western Blotting verification of three Caspase-3 siRNA, (E) Western Blotting verification of Casp-3 OE. * p < 0.05, ** p < 0.01, *** p < 0.001. n = 3 repeats.

after transfection (Fig 3B and 3C). The results showed that compared with the pcDNA3.1, the mRNA level of Caspase-3 in Casp3-OE group was significantly increased, and the difference between groups was statistically significant ($p < 0.001$). The protein band of Caspase-3 in Caspase-3 OE group was markedly thickened by WB at the same time, this indicated that the Casp3-siRNA-3 and Caspase-3 OE were successfully constructed and could be used for subsequent experiments (Fig 3D and 3E).

## 3.4 Dual luciferase reporter gene assay

The fluorescence intensity of let-7a-5p mimics + Caspase-3–3′UTR-WT was obviously weakened compared with that of NC mimics + Caspase-3–3′UTR- WT, and the difference between groups was statistically significant ($p < 0.001$). Compared with the NC mimics + Caspase-3–3′UTR-MUT, the fluorescence activity of let-7a-5p mimics + Caspase-3–3′UTR-MUT had no significant change (Fig 2B). The dual luciferase reporter gene activity assay confirmed mmu-let-7a-5p was targeted to the 3′UTR of Caspase-3, and Caspase-3 was one of the target genes of mmu-let-7a-5p.

## 3.5 BCG infection up-regulated the expression of mmu-let-7a-5p

qRT-PCR was performed to detect the expression levels of intracellular mmu-let-7a-5p and Caspase-3 at 24 h, 48 h and 72 h after BCG infection (Fig 4). Compared with the control group, the expression level of mmu-let-7a-5p in BCG group decreased slightly at 24 h time point, and the expression level of Caspase-3 was similar to that in the control group. There was no significant difference between the two groups. At 48 h after infection, compared with the control group, mmu-let-7a-5p was increased and Caspase-3 was decreased significantly in the BCG group, and the differences between groups were statistically significant ($p < 0.05$). At 72 h time point, the expression level of mmu-let-7a-5p and Caspase-3 had no significant differences between the two groups. It can be seen that BCG infection could up-regulate the expression of intracellular mmu-let-7a-5p, while the expression of Caspase-3 displayed the opposite trend.

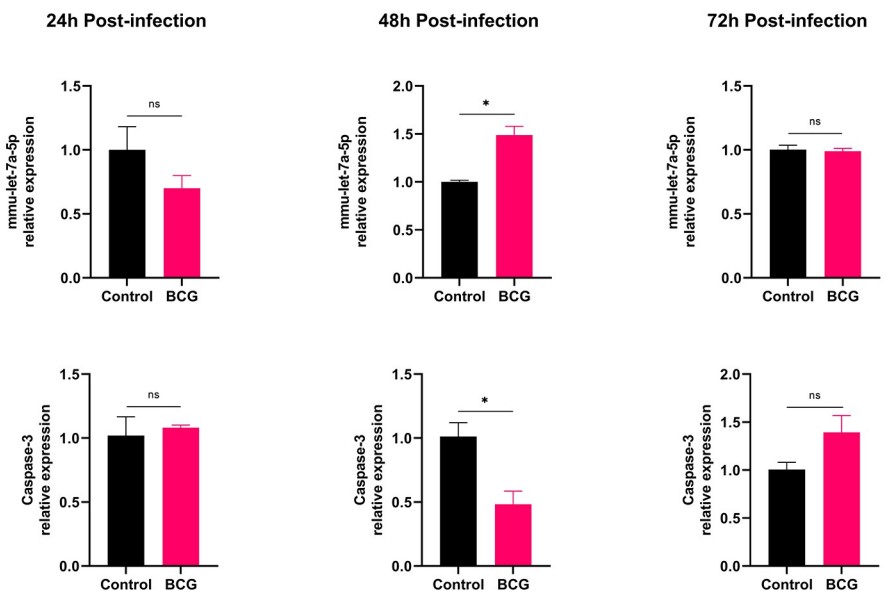

**Fig 4. The relative expression of mmu-let-7a-5p and Caspase-3 at different time points post BCG infection.** *
$p < 0.05$. n = 3 repeats.

### 3.6 Mmu-let-7a-5p negatively regulates Caspase-3

Meanwhile, the regulations of mmu-let-7a-5p on Caspase-3 were further observed by qRT-PCR and WB. Compared with the NC-mimics group, the mRNA level of Caspase-3 in the mmu-let-7a-5p mimics group was significantly decreased ($p < 0.01$); Compared with the NC-inhibitor group, the mRNA expression of Caspase-3 in the mmu-let-7a-5p inhibitor group was significantly increased ($p < 0.01$). The differences between the two groups were statistically significant. WB findings displayed that the protein expression changes of Caspase-3 were consistent with the result of qRT-PCR.

Furthermore, Raw264.7 macrophages were co-transfected with Casp3—OE and NC-mimics or let-7a-5p mimics, respectively. Compared with the NC mimics + Casp3—OE group, Caspase-3 expression level in the mmu-let-7a-5p mimics + Casp3—OE group was down-regulated, and the difference was statistically significant ($p < 0.001$). When Raw264.7 macrophages were co-transfected with Casp3—siRNA and NC-inhibitor or mmu-let-7a-5p inhibitor, separately, there was no significant difference between the two groups. The results of WB displayed that compared with NC mimics + Caspase-3 OE group, the protein band of Caspase-3 in mmu-let-7a-5p mimics + Caspase-3 OE group was apparently thinner; compared with the NC-inhibitor + Caspase-3—siRNA group, the protein strip of Capsase-3 in the mmu-let-7a-5p inhibitor + Caspase-3—siRNA group was significantly thickened. All of the above findings were presented in Fig 5.

### 3.7 Mmu-let-7a-5p enhances Raw264.7 cell viability

The effects of mmu-let-7a-5p and Caspase-3 on the viability of Raw264.7 cells were further clarified through CCK-8 assay. As shown in Fig 6, compared with the NC mimics group, the OD values of cells in the mmu-let-7a-5p-mimics group were significantly increased at each time points post infection, suggesting that the vitality of cells were enhanced and the difference between the two groups was statistically significant (Fig 6). Compared with the NC inhibitor

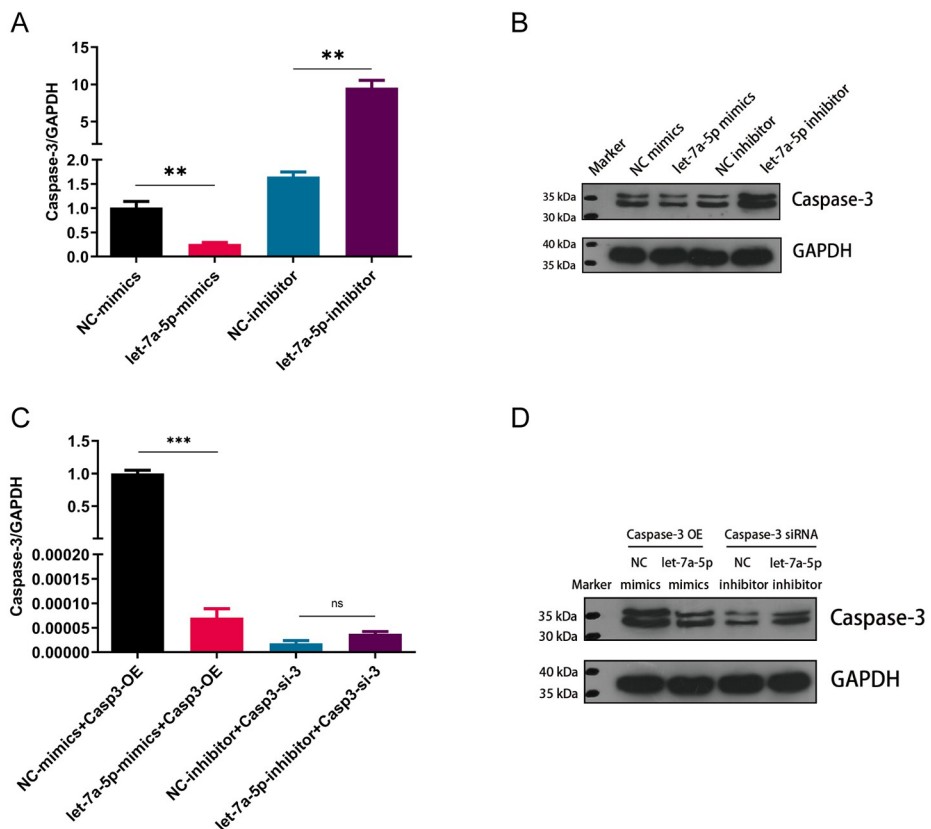

**Fig 5. Mmu-let-7a-5p negatively regulates Caspase-3 expression.** (A) The mRNA expression levels of Caspase-3 by qRT-PCR. (B) The relative protein expression levels of Caspase-3 by Western Blotting. (C) The mRNA expression changes of Caspase-3 in Rescue assay. (D) The relative protein expression changes of Caspase-3 in Rescue assay. ** $p < 0.01$, *** $p < 0.001$. n = 3 repeats.

group, the cell activity in the mmu-let-7a-5p-inhibitor group were decreased at 24 h ($p < 0.05$), significantly decreased at 48 h ($p < 0.01$), and slightly decreased at 72 h after infection, but the difference had no statistical significance (Fig 6B). Compared with the NC siRNA group, the activity of cells in the Casp3—siRNA-3 group were significantly increased at each time points after infection ($p < 0.01$) (Fig 6C). Compared with the pcDNA3.1 group, the cell vitality of cells was decreased in Casp3—OE group both at 24 h and 48 h t after infection. The difference between the two groups was statistically significant ($p < 0.05$), there was no difference between the two groups that has no statistical significance at 72 h time point (Fig 6D).

### 3.8 Mmu-let-7a-5p suppresses Raw264.7 cell apoptosis

In order to observe the effect on apoptosis regulated by mmu-let-7a-5p, Flow Cytometry was used to detect the apoptosis rate of Raw264.7 macrophages in different groups. As can be seen from the Fig 7A–7C, the average apoptosis rate of NC-mimics group was 29.60% ± 0.1922 and the mmu-let-7a-5p mimics group was 26.70% ± 0.367. Compared with NC-mimics group, the apoptosis rate in the mmu-let-7a-5p mimics group was significantly reduced and the difference between the two groups was statistically significant ($p < 0.01$). The average apoptosis rate of NC-inhibitor group was 28.24% ± 0.4683 and the mmu-let-7a-5p inhibitor group was 33.14% ± 0.5453. Compared with NC-inhibitor group, the apoptosis rate in the mmu-let-7a-5p inhibitor group was significantly increased ($p < 0.01$). In addition, the average apoptosis

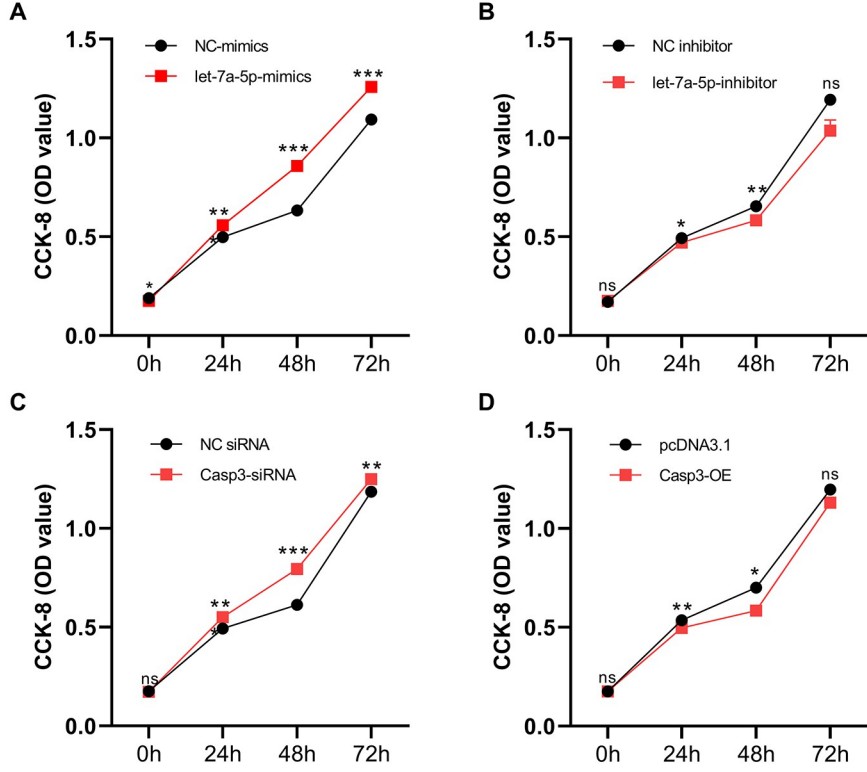

**Fig 6. Raw264.7 cell viability at different time points.** (A) NC mimics vs let-7a-5p mimics, (B) NC inhibitor vs let-7a-5p inhibitor, (C) NC siRNA vs Caspase-3 siRNA (Casp3-siRNA), (D) pcDNA3.1 vs Caspase-3 Over-expression plasmid (Casp-3-OE). * p < 0.05, ** p < 0.01, *** p < 0.001. n = 3 repeats.

rate of the NC-siRNA group was 36.03% ± 0.2226 and the Casp3—siRNA-3 group was 30% ± 0.374. Compared with the NC-siRNA group, the apoptosis rate in the Casp3—siRNA-3 group was decreased and the difference between these two groups was statistically significant ($p < 0.001$). The average apoptosis rate in pcDNA3.1 group was 73.83% ± 0.289 and the Casp3—OE group was 86.16% ± 0.3518. Compared with the pcDNA3.1 group, the apoptosis rate in the Casp3—OE group was obviously increased and the difference was also statistically significant ($p < 0.001$).

At the same time, one-way ANOVA was adopted to compare the apoptosis rates in the different groups. It was found in Fig 7D that compared with the Casp3—siRNA-3 group, the apoptosis rate in the mmu-let-7a-5p mimics group was decreased ($p < 0.01$), while significantly increased in the mmu-let-7a-5p inhibitor group ($p < 0.01$). Compared with the Casp3—siRNA-3 group, the apoptosis rate in the Casp3—OE group was significantly increased ($p < 0.001$). It can be seen that up-regulating the expression of mmu-let-7a-5p in Raw264.7 cells or silencing the expression of Caspase-3 could apparently suppress the apoptosis rate of macrophages, but down-regulating the expression of endogenous mmu-let-7a-5p or over-expressing the expression of intracellular Caspase-3 could increase the apoptosis rate of Raw264.7 by contrast.

### 3.9 BCG restrains TNFR1/FADD/Caspase-8/Caspse-3 signal axis

qRT-PCR was performed to detect the core genes in the extrinsic apoptotic signaling pathway mediated by TNFR1 (Fig 8A–8D). The consequences manifested that compared with the

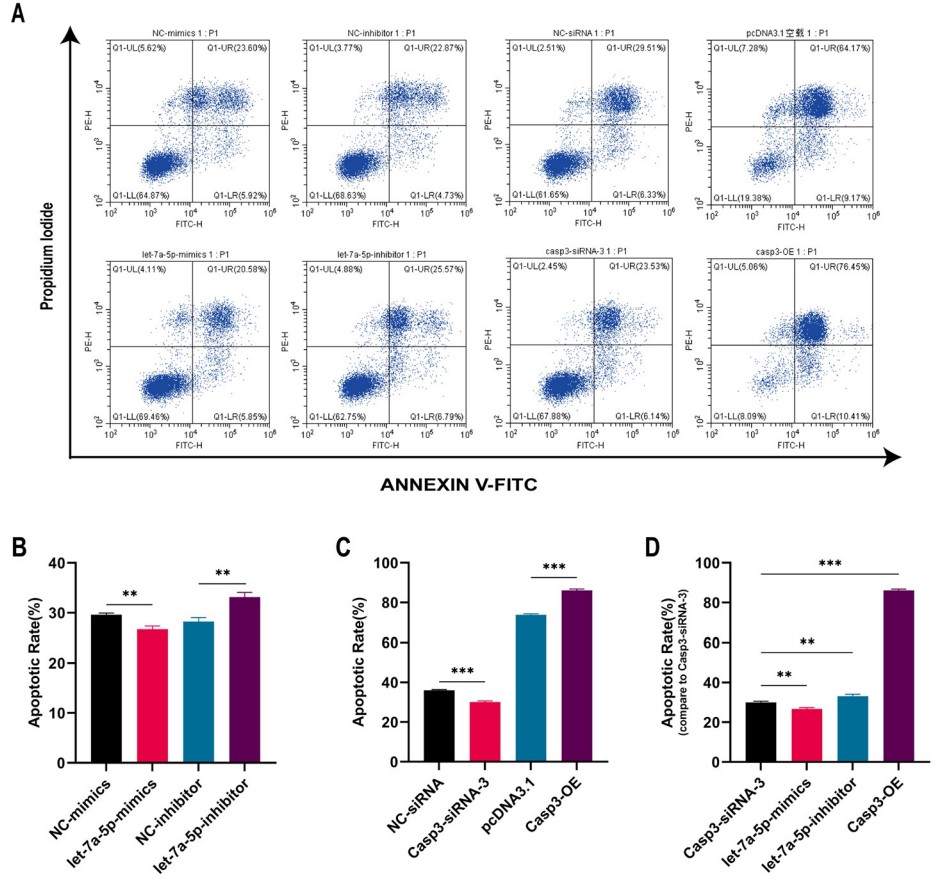

**Fig 7. Apoptosis rate of Raw264.7 cells in different groups at 48 hours after BCG infection.** (A) The results of Flow Cytometry, (B) Comparison of let-7a-5p mimics and inhibitor groups, (C) Comparison of Casp-3 siRNA and Casp-3 OE groups, (D) Comparison in multiple groups. ** $p < 0.01$, *** $p < 0.001$. n = 3 repeats.

control group, the expression level of TNFR1, FADD and Caspase-3 were decreased in the experiment groups, and the differences were statistically significant ($p < 0.05$). The expression level of Caspase-8 also showed a downward trend in the experiment group, but a slight rise in the let-7a-5p mimics + BCG group, however, there was no significant difference between the two groups compared with the control group. The comparative analysis of differences among multiple groups demonstrated that there were no significant differences in the expression of TNFR1 and FADD in let-7a-5p mimics + BCG and let-7a-5p inhibitor + BCG groups. The expression level of Caspase-8 in let-7a-5p inhibitor + BCG group was lower than that in let-7a-5p mimics + BCG group, and the difference was statistically significant ($p < 0.05$). The expression differences of Caspase-3 in different experiment groups were also statistically significant ($p < 0.001$). Moreover, the protein expression changes of TNFR1, Caspase-3 Pro and Caspase-3 cleaved were detected by Western Blotting (Fig 8E). The results displayed that all of the three protein expressions were increased in the let-7a-5p mimics / inhibitor + BCG group but decreased in the BCG group compared with the control group, and the differences between the two groups were statistically significant. It can be seen that compared with the BCG group, the expression changes of the above three proteins in let-7a-5p mimics + BCG group were similar with the let-7a-5p inhibitor + BCG group, and the differences between those groups were

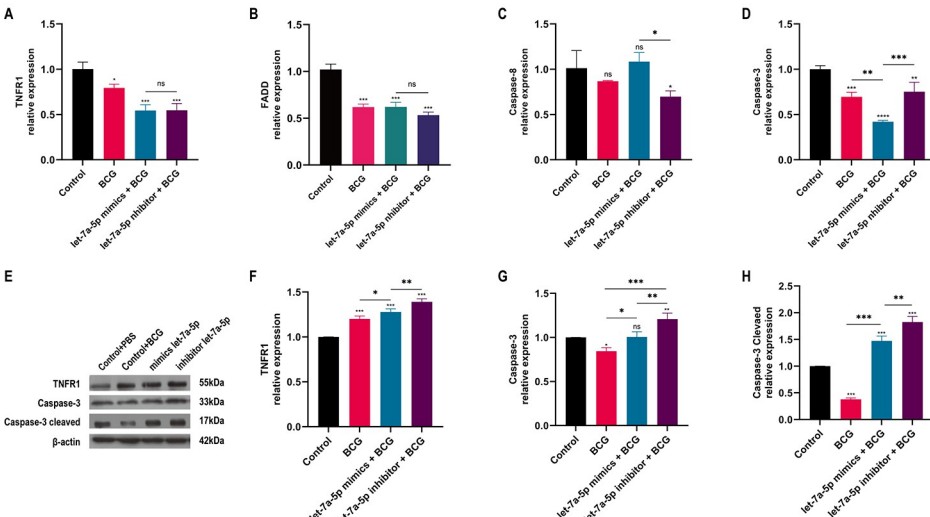

**Fig 8. The detection of core genes in TNFR1-mediated extrinsic apoptosis pathway.** (A-D) The mRNA expression levels of TNFR1, FADD, Caspase-8 and Caspase-3 by qRT-PCR, (E) The protein expression changes of TNFR1, Caspase-3 Pro and Caspase-3 cleaved by Western Blotting, (F-H) Quantitative analysis of TNFR1, Caspase-3 Pro and Caspase-3 cleaved protein expression in Western Blotting. * p < 0.05, ** p < 0.01, *** p < 0.001. n = 3 repeats.

statistically significant (Fig 8F–8H). Furthermore, it was found that the protein expression change of TNFR1 in the BCG group was opposite to the previous mRNA results, while the expression changes of Caspase-3 Pro and Caspase-3 cleaved in the BCG group were basically consistent with the mRNA results.

### 3.10 Caspase-3 siRNA inhibits TNF-α and IL-6 secretion

The expression levels of inflammatory cytokines at 48 hours after BCG infection were examined by ELISA. The results were presented in Fig 9. Compared with the control group, the secretion levels of TNF-α and IL-6 were significantly increased in the experimental groups except the Caspase-3 siRNA + BCG group, and the differences between the two groups were statistically significant ($p < 0.001$). The secretion of TNF-α and IL-6 was obviously decreased in the Caspase-3 siRNA + BCG group, and the difference between the two groups was also statistically significant. The expression of IL-1β and IFN-γ were not detected in each group. It indicated that Caspase-3 siRNA could inhibit the secretion of TNF-α and IL-6.

### 3.11 Mmu-let-7a-5p facilities BCG survival in Raw264.7 macrophage

The colony forming unit determination consequences displayed that compared with the control group treated with BCG stimulation alone, the CFU counts were significantly increased in the mmu-let-7a-5p mimics group and Caspase-3 siRNA group ($p < 0.01$) and decreased in the mmu-let-7a-5p inhibitor group and Caspase-3 OE group ($p < 0.05$), the differences were statistically significant (Fig 10).

## 4. Discussion

Studies have found that the fate of infected macrophages influenced the destiny of mycobacteria and the prognosis of tuberculosis infection to a certain degree. If macrophages underwent apoptosis, they could not only accelerate the progress in clearing intracellular mycobacteria at the early stage of infection, but also activate adaptive immune responses by antigen

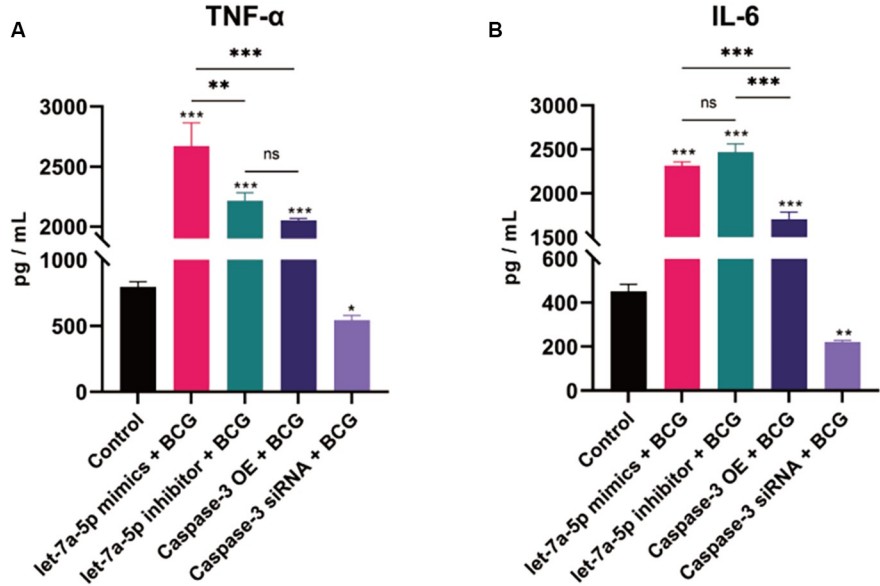

**Fig 9. The expression levels of inflammatory cytokines at 48 hours after BCG infection by ELISA.** (A) The expression changes of TNF-α in different groups, (B) The expression changes of IL-6 in different groups. * $p < 0.05$, ** $p < 0.01$, *** $p < 0.001$. n = 3 repeats.

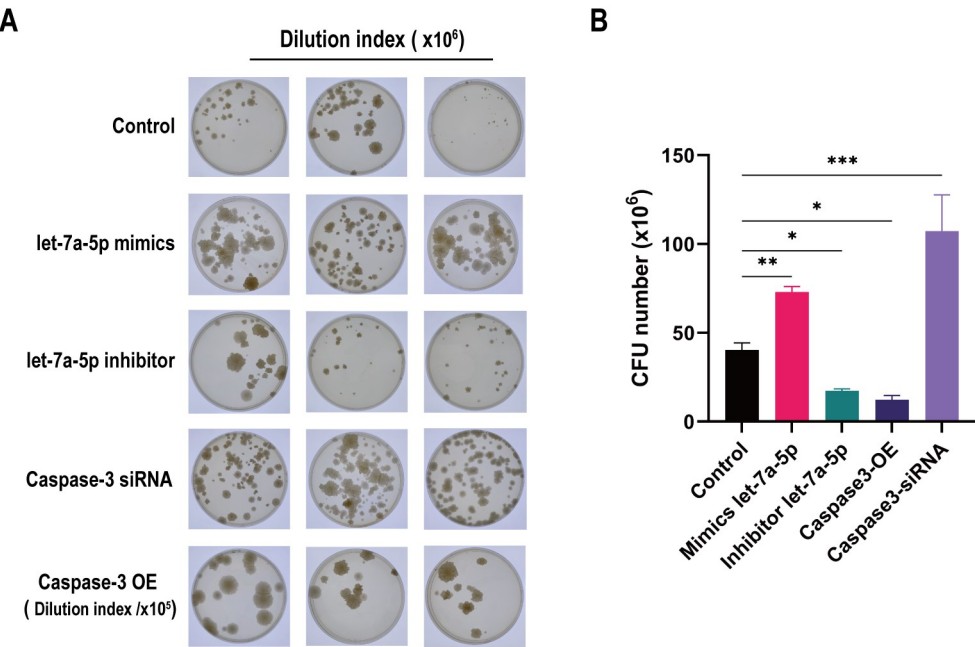

**Fig 10. The results of mycobacteria colony forming unit (CFU) determination.** (A) Mycobacteria colony forming unit counts in different groups after ×10⁶ dilution. (B) The CFU counts quantitative analysis in multiple groups. * $p < 0.05$, ** $p < 0.01$, *** $p < 0.001$. n = 3 repeats.

presentation to reduce bacterial load in the host [15–17, 21, 22]. Although a large number of studies have been reported on apoptosis pathways related to tuberculosis infection [23–26], the mechanism of how Mycobacterium tuberculosis regulates macrophages apoptosis remains unclear. The majority of previous studies explored the expression differences and functions of miRNA in macrophages post Mycobacterium tuberculosis infection from the cellular level, while we attempted to investigate from the exosome level in our previous study [18]. We found that BCG could change the profile of miRNA in exosomes derived from the infection group cell culture supernatants. The expression of mmu-let-7a-5p and other five miRNAs were up-regulated in the infection group. Bioinformatics analysis indicated that mmu-let-7a-5p and its target genes were involved in multiple biological processes and pathways and correlated with the regulation of apoptosis. Similar findings were reported, Aplipoor et al. [27] infected human monocyte-derived macrophages with BCG and identified a complex set of exosomal miRNAs. The enrichment analysis illustrated that they were primarily taken part in the regulation of host metabolic progression which participate in protective immunity of the host, and this effect ultimately leads to the persistence of intracellular bacteria. Mtb is a facultative intracellular parasite. Macrophages were the first barrier to defend the Mtb invasion, at the same time, they can be utilized as a potential tent by Mtb to induce latent infections [28–30]. There is evidence that bacteria could reverse the apoptosis process of macrophages by manipulating host miRNAs to achieve its immune escape and persistent survival [31, 32], but the specific mechanisms are still unclear. Macrophage apoptosis has been considered as an effective protecting mechanism for controlling tuberculosis infection and plays a key role in the clearance of intracellular Mycobacterium tuberculosis for a long time [26].

Let-7 is the second miRNA discovered after lin-4, which is extremely conserved in species including humans [33]. Since its discovery, let-7 has expanded greatly and paved the way for non-coding RNA (ncRNA) research. It has been found that let-7 family members are important regulatory factors that participate in the regulation of various human diseases [33–36]. Previous studies on let-7 family mainly focus on cancer and viral infection. Tsang et al. first reported in 2008 that let-7a can affect drug-induced apoptosis of A431 cells and HepG2 cells through Caspase-3; Fasihi-Ramandi et al. found in 2018 that let-7a-5p could block Caspase-3 to induce apoptosis of HL60 cells and inhibit the proliferation of AML [37]. However, there were few studies on let-7a-5p and Caspase-3 in tuberculosis infection.

The bioinformatics analysis and the validation of dual luciferase reporter gene assay confirmed that Caspase-3 was one of the target genes of mmu-let-7a-5p. We examined the intracellular miRNA expression changes at different time points after BCG infection. The results showed that compared with the control group, the expression level of intracellular mmu-let-7a-5p fluctuated with time after BCG infection, showing an overall upward trend. The expression of intracellular mmu-let-7a-5p increased significantly at 48 h after BCG infection, which was consistent with the results of high-throughput sequence. In the meantime, the expression of Caspase-3 showed the opposite trend with mmu-let-7a-5p, this indicated that BCG infection could up-regulate the expression of mmu-let-7a-5p in Raw264.7 cells. However, whether mmu-let-7a-5p could inhibit the apoptosis of macrophages by targeting Caspase-3, which was beneficial to the survival of mycobacteria in the cell and escape immune clearance needed to be further explored.

Apoptosis is a programmed cell death by activating Caspase to clear damaged or multiple cells [38, 39]. Cysteine Aspartate Acid Specific Proteases (Caspase) are a family of proteases closely related to the regulation of apoptosis [40, 41]. They are divided into Initiators (Caspase-2 / -8 / -9 / -10) and Effectors (Caspase-3 / -6 / -7) according to the presence or absence of specific N-terminal interaction domain proteins. Caspase-3 is a key effector protein kinase,

which needs to be activated under the action of other family members. Once the enzyme is activated, irreversible programmed cell death will occur [42].

CCK-8 assay and flow cytometry were utilized to detect the proliferation ability, viability and apoptosis rate of Raw264.7 cells in different groups. Up- regulating the expression of mmu-let-7a-5p in cells or silencing Caspase-3 significantly enhanced the proliferation ability and viability and decreased the apoptosis rate of the cells, while down-regulating of mmu-let-7a-5p or over-expressing of Caspase-3 had the contrary consequence on the cell proliferation and cell viability even the apoptosis rate. In other words, mmu-let-7a-5p negatively regulates the apoptosis of Raw264.7 cells, whilst Caspase-3 positively regulates the apoptosis of Raw264.7 cells. Rescue experiment confirmed that mmu-let-7a-5p could attenuate the biological effects caused by Caspase-3 OE to a certain extent, which indicated that mmu-let-7a-5p could directly and negatively regulate Caspase-3 expression.

We detected the expression changes of inflammatory factors TNF-α, IL-6, IL-1β and IFN-γ in the supernatant of cells in each group at the time point of 48 h after BCG infection. Compared with the control group, the expressions of TNF-α and IL-6 were significantly increased in the experimental groups except the Caspase-3 siRNA group and the expression of IL-1β and IFN-γ could not be detected at the same time. Overexpression of mmu-let-7a-5p could significantly increase the secretion of TNF-α while inhibiting the expression of mmu-let -7a-5p could obviously increase the secretion of IL-6. The expression of Caspase-3 had a certain correlation with the secretion of TNF-α and IL-6. Silencing the expression of Caspase-3 was capable of inhibiting the production of TNF-α and IL-6, but it cannot completely rule out the influence of Off-Target effect caused by cell transfection reagent and other factors on the function of Raw264.7 cells.

As a key effector of apoptosis, the activation of Caspases mainly mediated extrinsic and intrinsic apoptosis [43, 44]. Mitochondrial outer membrane permeability (MOMP) changes and cytochrome C release are the central events of this pathway, followed by apoptotic protease activation factor (Apaf-1) combined with cytochrome C to form apoptotic bodies to further activate Caspase-9. The extrinsic pathway stimulates the TNF receptor subfamily members on the cell membrane surface, such as TNFRI, Fas or TRAILR by specific ligands such as TNF-α, FasL or TRAIL, and then initiates, which further mediates the recruitment and activation of Caspase-8 / -10 through DISC. DISC is a death-induced signaling complex composed of Fas-related death domain protein (FADD) and / or TNFR-related death domain protein (TRADD) [45]. No matter which apoptotic pathway Caspases are activated, they will eventually mediate the apoptotic effect executor Caspase-3 activation to initiate apoptosis. Perforin and granzyme B are important proteins released by cytotoxic lymphocytes such as CTL and NK [46, 47]. They also directly initiate and activate Caspase to induce apoptosis.

In the process of viral and bacterial infections infection, extrinsic apoptosis initiated by the binding of TNFR-specific ligand TNF-α to cell surface TNFR1 is a way for host cells to clear mycobacterial pathogens [48–51]. We detected the expression changes of key genes of TNFR1/ FADD/Caspase-8/Caspase-3 signaling axis in this apoptosis pathway by qRT-PCR. Compared with the control group, the expression of TNFR1, FADD, Caspase-8 and Caspase-3 were significantly down-regulated in the BCG groups, indicating that the TNFR1/FADD/Caspase-8/ Caspase-3 signaling axis of extrinsic apoptosis mediated by TNFR1 was inhibited, which also explained to some extent that BCG infection could inhibit the apoptosis of macrophages to avoid being cleared by macrophages and be beneficial to their persistence in the host cells. In addition, the changes of mmu-let-7a-5p expression level in Raw264.7 macrophages had no significant effect on the expression of TNFR1 and FADD, and the expression of Caspase-8 in let-7a-5p inhibitor group was lower than that in let-7a-5p mimics group, while Caspase-3 was significantly correlated with the expression changes of intracellular let-7a-5p. These results also

certificated that mmu-let-7a-5p could directly and negatively regulate the key executive mole-cule Caspase- 3 expressions during the apoptosis.

Generally, Caspase-3 has two forms in vivo, one is an inactive zymogen, the full-length Cas-pase-3 (Caspase-3 Pro), and the other is an active activated Caspase-3 (Caspase-3 cleaved) [40, 42]. The Caspase-3 originally has two restriction sites, and the activation of the full-length Cas-pase-3 requires cleavage at these two restriction sites to generate the biologically active acti-vated Caspase-3 fragment (17 kDa) and inactive fragments. To our knowledge, the detection of the activated Caspase-3 fragment at the protein level should be an accurate indicator for proving the occurrence of apoptosis. Therefore, we detected the protein expression changes of Caspase-3 clevaed, Caspase-3 Pro and TNFR1 by Western Blot. Compared with the control group, all of the three proteins were obviously increased in the BCG + let-7a-5p mimic / inhib-itor groups. Combined with the qRT-PCR results, we found that the protein expression of TNFR1 were different from the mRNA results. We could not completely explain the real rea-son behind this phenomenon, however, it is certain that the protein expression of Caspase-3 Pro and Caspase-3 cleaved in the BCG group was decreased compared with the control group, and the decrease of Caspase-3 cleaved was more significant. This was basically consistent with the qPCR results of mRNA, which implied that BCG infection can inhibit the apoptosis of macrophages. It also provided a relatively strong theoretical basis for elucidating the mecha-nism of immune escape of Mycobacterium.

Finally, we estimated the intracellular bacterial load and clearance rate of mycobacteria by the determination of colony formation unit. The results showed that mmu-let-7a-5p positively regulated the intracellular bacterial load and facilitated its survival, while Caspase-3 negatively modulated the intracellular bacterial load and was beneficial to the clearance of mycobacteria in macrophages.

## 5. Conclusion

Mmu-let-7a-5p negatively regulates the apoptosis of Raw264.7 macrophages by targeting Cas-pase-3. Down-regulated the expression of mmu-let-7a-5p facilities the clearance of residual mycobacteria and attenuation the bacterial load of intracellular mycobacteria. Therefore, mmu-let-7a-5p has the potential to be a target for anti-tuberculosis immunotherapy. In our subsequent study, we will expect to develop a novel exosomal drug system for synergistic deliv-ery of mmu-let-7a-5p and rifampicin for the treatment of bone and joint tuberculosis.

## Supporting information

**S1 Raw images. The data including original uncropped and unadjusted blot/gel images of this manuscript.**
(PDF)

## Author Contributions

**Conceptualization:** Rong Ma, Zhaohui Ge.

**Methodology:** Xuehua Zhan.

**Software:** Wenqi Yuan.

**Supervision:** Guangxian Xu.

**Validation:** Xuehua Zhan, Wenqi Yuan, Yueyong Zhou.

**Writing – original draft:** Xuehua Zhan.

**Writing – review & editing:** Zhaohui Ge.

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
