## [Decision Letter · Decision Letter 0]

9 Jan 2024

PONE-D-23-31553mmu-let-7a-5p inhibits macrophage apoptosis by targeting CASP3 to increase bacterial load and promote facilities mycobacteria survivalPLOS ONE

Dear Dr. Ge,

Thank you for submitting your manuscript to PLOS ONE. After careful consideration, we feel that it has merit but does not fully meet PLOS ONE’s publication criteria as it currently stands. Therefore, we invite you to submit a revised version of the manuscript that addresses the points raised during the review process.

We look forward to receiving your revised manuscript.

Kind regards,

Francesco Sessa, Ph.D., MS

Academic Editor

PLOS ONE

5. PLOS requires an ORCID iD for the corresponding author in Editorial Manager on papers submitted after December 6th, 2016. Please ensure that you have an ORCID iD and that it is validated in Editorial Manager. To do this, go to ‘Update my Information’ (in the upper left-hand corner of the main menu), and click on the Fetch/Validate link next to the ORCID field. This will take you to the ORCID site and allow you to create a new iD or authenticate a pre-existing iD in Editorial Manager. Please see the following video for instructions on linking an ORCID iD to your Editorial Manager account: https://www.youtube.com/watch?v=_xcclfuvtxQ.

6. Please amend either the title on the online submission form (via Edit Submission) or the title in the manuscript so that they are identical.

7. PLOS ONE now requires that authors provide the original uncropped and unadjusted images underlying all blot or gel results reported in a submission’s figures or Supporting Information files. This policy and the journal’s other requirements for blot/gel reporting and figure preparation are described in detail at https://journals.plos.org/plosone/s/figures#loc-blot-and-gel-reporting-requirements and https://journals.plos.org/plosone/s/figures#loc-preparing-figures-from-image-files. When you submit your revised manuscript, please ensure that your figures adhere fully to these guidelines and provide the original underlying images for all blot or gel data reported in your submission. See the following link for instructions on providing the original image data: https://journals.plos.org/plosone/s/figures#loc-original-images-for-blots-and-gels.

Additional Editor Comments:

This study aims to explore the impact of mmu-let-7a-5p on Raw264.7 mononuclear macrophage apoptosis and its influence on intracellular Bacillus Calmette-Guérin (BCG) load in vitro. While the study holds potential interest for readers, there are several significant changes required.

The abstract section needs enhancement, specifically in introducing the theme, outlining the aims, and summarizing methods and results. Additionally, the conclusion section should outline future research directions, as the current presentation is unclear.

The introduction section should also be improved with background information relating to the study's aims. Furthermore, the material and methods section would benefit from the inclusion of a schematic diagram illustrating the study's protocol and a justification for using GAPDH as a reference gene for qPCR detection of mRNA. Additionally, the authors should provide the original figures of the western blot as supplementary files for the results section and submit the main raw data as supplementary files.

In the discussion section, the authors should refrain from comparing the present study's data to their previous findings and expand the analysis to include international data. Moreover, it should be important to insert the section limitations.

Lastly, the conclusion section should suggest future research directions.

Minor points:

- English should be reviewed by a native speaker.

- Please, check the use of reference manager software (Error! Reference source not found..-> line 121).

- The short form should be introduced by the extension form the first time; subsequently, it should be used through the text. Please, check all abbreviations.

Reviewers' comments:

Reviewer's Responses to Questions

**Comments to the Author**

1. Is the manuscript technically sound, and do the data support the conclusions?

Reviewer #1: No

Reviewer #2: Partly

2. Has the statistical analysis been performed appropriately and rigorously? 

Reviewer #1: I Don't Know

Reviewer #2: N/A

3. Have the authors made all data underlying the findings in their manuscript fully available?

Reviewer #1: Yes

Reviewer #2: Yes

4. Is the manuscript presented in an intelligible fashion and written in standard English?

Reviewer #1: No

Reviewer #2: Yes

5. Review Comments to the Author

Reviewer #1: I am sorry that I have little understanding of the content of this article and cannot make scientific and correct judgments about it. Please find another reviewer. I'm sorry that I clicked on 'accept review' earlier (before seeing the manuscript) because it caused me to review the manuscript without being familiar with it, which is unfair to the author.

Reviewer #2: This study found that mmu-let-7a-5p regulates the mycobacterial load in macrophages through apoptosis, which is of certain value and innovation, because in addition, reports about it focus on tumors and cardiovascular diseases, and few reports are related to infectious diseases. But there are some issues in the article that need to be revised

1. The figure legends were not clear, and there was no explanation of the production method or number of replicates for each group of samples.

2. There are A-H graphs in Figure 8, but only A-B is explained and does not match

3. There are some columnar statistical charts, although the p-value is less than 0.05, the actual difference is not significant, which usually means that there may be some deviation between the statistical significance and the actual biological significance. As shown in Figure 8F, Figure 7B, etc

4. Figure 8G and Figure 8H, qRT PCR to detect mRNA of Caspase-3 Pro and Caspase-3 cleared. In fact, qRT PCR cannot distinguish between Caspase-3 Pro and Caspase-3 cleared. You need to explain.

6. PLOS authors have the option to publish the peer review history of their article (what does this mean?). If published, this will include your full peer review and any attached files.

Reviewer #1: No

Reviewer #2: No

---

## [Author Response · Author response to Decision Letter 0]

19 Mar 2024

Dear Editors and Reviewers:

Thank you for your letter and for the reviewer’s comments concerning our manuscript entitled “mmu-let-7a-5p inhibits macrophage apoptosis by targeting CASP3 to increase bacterial load and promote facilities mycobacteria survival” [manuscript number: PONE-D-23-31553]. 

First of all, please let me express my respect and sincere thanks for your conscientious work attitude. Those comments are all valuable and very helpful for revising and improving our manuscript, as well as the important guiding significance to our researchers. 

Review comments

This study found that mmu-let-7a-5p regulates the mycobacterial load in macrophages through apoptosis, which is of certain value and innovation, because in addition, reports about it focus on tumors and cardiovascular diseases, and few reports are related to infectious diseases. But there are some issues in the article that need to be revised

1. The figure legends were not clear, and there was no explanation of the production method or number of replicates for each group of samples.

2. There are A-H graphs in Figure 8, but only A-B is explained and does not match

3. There are some columnar statistical charts, although the p-value is less than 0.05, the actual difference is not significant, which usually means that there may be some deviation between the statistical significance and the actual biological significance. As shown in Figure 8F, Figure 7B, etc

4. Figure 8G and Figure 8H, qRT PCR to detect mRNA of Caspase-3 Pro and Caspase-3 cleared. In fact, qRT PCR cannot distinguish between Caspase-3 Pro and Caspase-3 cleared. You need to explain.

Reply

We have studied comments carefully and have made correction which we hope meet with approval. Revised portion are marked in red in the paper. The main corrections in the paper and the responds to the reviewer’s comments are as flowing:

1. The figure legends were not clear, and there was no explanation of the production method or number of replicates for each group of samples.

Thank you for your comments. We have revised our figure legends and annotated description of replicates for each group of samples.

2. There are A-H graphs in Figure 8, but only A-B is explained and does not match

We are very sorry for our negligence of these mistakes in the manuscript writing and we have carefully revised these mistakes.

3. There are some columnar statistical charts, although the p-value is less than 0.05, the actual difference is not significant, which usually means that there may be some deviation between the statistical significance and the actual biological significance. As shown in Figure 8F, Figure 7B, etc.

We are extremely grateful to Reviewer for pointing out this problem. We have double checked the raw data and re-calculated the statistics by using Graphpad Prism 9.0. There are indeed significant differences in the results of statistical charts in our manuscripts.

4. Figure 8G and Figure 8H, qRT PCR to detect mRNA of Caspase-3 Pro and Caspase-3 cleared. In fact, qRT PCR cannot distinguish between Caspase-3 Pro and Caspase-3 cleared. You need to explain.

Due to our negligence, we did not explain exactly about the figure legends, especially Figure 8. In fact, Figure 8.F-H were the quantitative analysis of TNFR1, Caspase-3 Pro and Caspase-3 cleaved protein expression in Western Blotting, they were not the expression level of mRNA.

Additional Editor Comments

This study aims to explore the impact of mmu-let-7a-5p on Raw264.7 mononuclear macrophage apoptosis and its influence on intracellular Bacillus Calmette-Guérin (BCG) load in vitro. While the study holds potential interest for readers, there are several significant changes required.

1. The abstract section needs enhancement, specifically in introducing the theme, outlining the aims, and summarizing methods and results. Additionally, the conclusion section should outline future research directions, as the current presentation is unclear.

2. The introduction section should also be improved with background information relating to the study's aims. Furthermore, the material and methods section would benefit from the inclusion of a schematic diagram illustrating the study's protocol and a justification for using GAPDH as a reference gene for qPCR detection of mRNA. Additionally, the authors should provide the original figures of the western blot as supplementary files for the results section and submit the main raw data as supplementary files.

3. In the discussion section, the authors should refrain from comparing the present study's data to their previous findings and expand the analysis to include international data. Moreover, it should be important to insert the section limitations.

4. Lastly, the conclusion section should suggest future research directions.

Reply

1. The abstract section needs enhancement, specifically in introducing the theme, outlining the aims, and summarizing methods and results. Additionally, the conclusion section should outline future research directions, as the current presentation is unclear.

Thank you for your comments. Considering the Editor’s suggestion, we have re-written the abstract section part. 

Abstract: We have been trying to find a miRNA that can specifically regulate the function of mycobacterial host cells to achieve the purpose of eliminating Mycobacterium tuberculosis. The purpose of this study is to investigate the regulation of mmu-let-7a-5p on macrophages apoptosis and its effect on intracellular BCG clearance. After a series of in vitro experiments, we found that mmu-let-7a-5p could negatively regulate the apoptosis of macrophages. by targeting Caspase-3. The extrinsic apoptosis signal axis TNFR1/FADD/Caspase-8/Caspase-3 was inhibited after BCG infection. Up-regulated the expression level of mmu-let-7a-5p increase the cell proliferation viability and inhibit apoptosis rate of macrophages, but down-regulated its level could apparently reduce the bacterial load of intracellular Mycobacteria and accelerate the clearance of residual Mycobacteria effectively. Mmu-let-7a-5p has great potential to be utilized as an optimal candidate exosomal loaded miRNA for anti-tuberculosis immunotherapy in our subsequent research.

2. The introduction section should also be improved with background information relating to the study's aims. Furthermore, the material and methods section would benefit from the inclusion of a schematic diagram illustrating the study's protocol and a justification for using GAPDH as a reference gene for qPCR detection of mRNA. Additionally, the authors should provide the original figures of the western blot as supplementary files for the results section and submit the main raw data as supplementary files.

Thank you for your comments and we have updated the background information of the introduction section. Furthermore, a schematic diagram was supplied to the material and methods.

Glyceraldehyde-3-phosphate Dehydrogenase (GAPDH) is widely present in many organisms and is abundant in cells, accounting for 10% -20% of the total protein. The gene has a highly conserved sequence, which is highly expressed in almost all tissues. The protein expression in the same cell or tissue is generally constant, and it is not affected by the induction substances such as partial recognition check point and phorbol lipid. It has been commonly used as a reference gene in RT-qPCR assays for gene and protein expression analysis. Due to this, we chose GAPDH as a reference gene for qPCR detection of mRNA.

Additionally, we have re-submitted the original figures of the western blot as supplementary files for the results section and submit the main raw data as supplementary files.

3. In the discussion section, the authors should refrain from comparing the present study's data to their previous findings and expand the analysis to include international data. Moreover, it should be important to insert the section limitations.

Thank you for your comments. We have modified the discussion section according to your comments.

4. The conclusion section should suggest future research directions.

Thank you for your comments. Based on our primary research, we will expect to construct a new multiple drug delivery system with exosomes for tuberculosis treatment in the future, which loaded miRNA and traditional anti-tuberculosis drug rifampin. This was mentioned in the conculsion.

Conclusion: Mmu-let-7a-5p negatively regulates the apoptosis of Raw264.7 macrophages by targeting Caspase-3. Down-regulated the expression of mmu-let-7a-5p facilities the clearance of residual mycobacteria and attenuation the bacterial load of intracellular mycobacteria. Therefore, mmu-let-7a-5p has the great potential to be utilized as a candidate miRNA for the construction of novel exosomal drug delivery system loading with rifampin for anti-tuberculosis therapy in our subsequent research.

Minor points:

- English should be reviewed by a native speaker.

- Please, check the use of reference manager software (Error! Reference source not found..-> line 121).

- The short form should be introduced by the extension form the first time; subsequently, it should be used through the text. Please, check all abbreviations.

Reply

1. English should be reviewed by a native speaker.

Thank you for your advice. We not only invited an English-speaking friend from the United States to help polish our manuscript. Our manuscripts are also edited by one or more highly qualified native English editors of AJE for appropriate English language, grammar, punctuation, spelling and overall style.

2. Please, check the use of reference manager software (Error! Reference source not found.-> line 121).

Thank you for your comments. We are very sorry for our negligence of the mistake in line 121 of the manuscript and we have revised the mistake.

3. The short form should be introduced by the extension form the first time; subsequently, it should be used through the text. Please, check all abbreviations.

According to the Reviewer’s suggestion, we have double checked all the abbreviations in our manuscript.

We tried our best to improve the quality of our manuscript and made some changes to the manuscript. These changes will not influence the content and framework of the paper. And we not only listed the changes in the email but also marked in red in the revised manuscript. We appreciate Editors / Reviewers’ warm work earnestly, and hope that the correction will meet with approval.

---

## [Decision Letter · Decision Letter 1]

22 May 2024

PONE-D-23-31553R1Mmu-let-7a-5p inhibits macrophage apoptosis by targeting CASP3 to increase bacterial load and facilities mycobacterium survivalPLOS ONE

Dear Dr. Ge,

Thank you for submitting your manuscript to PLOS ONE. After careful consideration, we feel that it has merit but does not fully meet PLOS ONE’s publication criteria as it currently stands. Therefore, we invite you to submit a revised version of the manuscript that addresses the points raised during the review process.

We look forward to receiving your revised manuscript.

Kind regards,

Francesco Sessa, Ph.D., MS

Academic Editor

PLOS ONE

Journal Requirements:

Additional Editor Comments:

The reviewer suggested minor revision and I agree with his comments.

Reviewers' comments:

Reviewer's Responses to Questions

**Comments to the Author**

1. If the authors have adequately addressed your comments raised in a previous round of review and you feel that this manuscript is now acceptable for publication, you may indicate that here to bypass the “Comments to the Author” section, enter your conflict of interest statement in the “Confidential to Editor” section, and submit your "Accept" recommendation.

Reviewer #3: (No Response)

2. Is the manuscript technically sound, and do the data support the conclusions?

Reviewer #3: Yes

3. Has the statistical analysis been performed appropriately and rigorously? 

Reviewer #3: Yes

4. Have the authors made all data underlying the findings in their manuscript fully available?

Reviewer #3: Yes

5. Is the manuscript presented in an intelligible fashion and written in standard English?

Reviewer #3: Yes

6. Review Comments to the Author

**Reviewer #3: **The re-submitted manuscript entitled “Mmu-let-7a-5p inhibits macrophage apoptosis by targeting CASP3 to increase bacterial load and facilities mycobacterium survival” by Zhan et al have made considerable modifications, however, there are still some minor concerns that need further addressed before accept.

1. Line 253 “3.7 Mmu-let-7a-5p enhances Raw264.7 cell proliferation and viability”, according to Fig 6 and its-related contents, the current subtitle of 3.7 needs to rethink and refine, especially only CCK-8 results whether could support the cell proliferation as authors claim?

2. Figure 7, if possible, suggests the authors add a group that is just Raw264.7 cells without anything as a negative control. The current apoptotic data from Nc-mimics and NC-siRNA almost reached 30%, whether the Nc-mimics and NC-siRNA could promote apoptosis? To exclude the possibility of higher apoptotic rates such as Nc-mimics and NC-siRNA groups, the cell picture from each should provide additional support.

3. In Fig 10, it’s better to add the dilution index on the top of panel A.

4. Considering the large amounts of data and results of this study, a graphic abstract is helpful if it is provided.

7. PLOS authors have the option to publish the peer review history of their article (what does this mean?). If published, this will include your full peer review and any attached files.

Reviewer #3: No

---

## [Author Response · Author response to Decision Letter 1]

15 Jul 2024

Response to Reviewers

Dear Editors and Reviewers,

Thank you for your response and for the reviewer’s comments concerning our manuscript entitled “Mmu-let-7a-5p inhibits macrophage apoptosis by targeting CASP3 to increase bacterial load and promote facilities mycobacteria survival” [manuscript number: PONE-D-23-31553R1]. 

Please let me express my respect and sincere thanks for your conscientious work attitude. Those comments are all valuable and very helpful for revising and improving our manuscript, as well as the important guiding significance to our researchers. 

Journal Requirements

Reply

We have reviewed our references and double-checked the references list, no article was retracted.

Review comments

Reviewer #3: The re-submitted manuscript entitled “Mmu-let-7a-5p inhibits macrophage apoptosis by targeting CASP3 to increase bacterial load and facilities mycobacterium survival” by Zhan et al. have made considerable modifications; however, there are still some minor concerns that need further addressed before accepted.

1. Line 253 “3.7 Mmu-let-7a-5p enhances Raw264.7 cell proliferation and viability”, according to Fig. 6 and its related contents, the current subtitle of 3.7 needs to rethink and refine, especially only CCK-8 results whether could support the cell proliferation as authors claim?

2. Figure 7, if possible, suggests the authors add a group that is just Raw264.7 cells without anything as a negative control. The current apoptotic data from Nc-mimics and NC-siRNA almost reached 30%, whether the Nc-mimics and NC-siRNA could promote apoptosis? To exclude the possibility of higher apoptotic rates such as Nc-mimics and NC-siRNA groups, the cell picture from each should provide additional support.

3. In Fig. 10, it is better to add the dilution index on the top of panel A.

4. Considering the large amounts of data and results of this study, a graphic abstract is helpful if it is provided.

Reply

We have studied comments carefully and have made corrections which we hope meet with approval. Revised portions are marked in red in the paper. The main corrections in the paper and the responses to the reviewer’s comments are as follows:

1. Line 253 “3.7 Mmu-let-7a-5p enhances Raw264.7 cell proliferation and viability”, according to Fig. 6 and its related contents, the current subtitle of 3.7 needs to rethink and refine, especially only CCK-8 results whether could support the cell proliferation as authors claim?

Thank you for your comments and we are extremely grateful to the Reviewer for pointing out this problem. Considering the reviewer’s comment, we have revised our subtitle of 3.7 and figure legends of Fig. 6. The CCK-8 results are appropriate to demonstrate the cell viability.

2. Figure 7, if possible, suggests the authors add a group that is just Raw264.7 cells without anything as a negative control. The current apoptotic data from Nc-mimics and NC-siRNA almost reached 30%, whether the Nc-mimics and NC-siRNA could promote apoptosis? To exclude the possibility of higher apoptotic rates such as Nc-mimics and NC-siRNA groups, the cell picture from each should provide additional support.

First of all, we thank you for your rigorous and realistic academic style. In our data, there is a blank cell group which is Raw264.7 cells without anything as a negative control as you referred to above. Because the processing of apoptosis data needs to be compared with blank cells, the histogram shown later is actually the value compared with blank cells. Although the apoptosis rate of the Nc-mimics group and the NC-siRNA group was nearly 30%, they were significantly different from the statistical analysis results between the let-7a-5p mimics group and the Casp-3 siRNA group.

3. In Fig. 10, it is better to add the dilution index on the top of panel A.

Thank you for your comments. Considering the reviewer’s suggestion, we have added the dilution index on the top of panel A.

4. Considering the large amounts of data and results of this study, a graphic abstract is helpful if it is provided.

Thank you so much for your affirmation and suggestions for our work. In fact, we have already prepared a graphic abstract before our first submission. We will upload the graphic abstract together this time. Please review and make clear whether it is suitable for publication.

Finally, we sincerely thank the reviewers for their valuable comments and suggestions. We try our best to revise the manuscript and improve the quality of our manuscripts by making some modifications. These changes will not affect the content and framework of the paper. We sincerely thank the editors / reviewers for their enthusiastic work and hope that the review work will be recognized.

Yours sincerely

Zhaohui Ge

---

## [Editor Report · Decision Letter 2]

17 Jul 2024

Mmu-let-7a-5p inhibits macrophage apoptosis by targeting CASP3 to increase bacterial load and facilities mycobacterium survival

PONE-D-23-31553R2

Dear Dr. Ge,

We’re pleased to inform you that your manuscript has been judged scientifically suitable for publication and will be formally accepted for publication once it meets all outstanding technical requirements.

Kind regards,

Francesco Sessa, Ph.D., MS

Academic Editor

PLOS ONE

Additional Editor Comments (optional):

Following the reviewers' comments, the authors have improved their manuscript. I endorse its publication.
---

## [Editor Report · Acceptance letter]

24 Jul 2024

PONE-D-23-31553R2 

PLOS ONE

Dear Dr. Ge, 

I'm pleased to inform you that your manuscript has been deemed suitable for publication in PLOS ONE. Congratulations! Your manuscript is now being handed over to our production team.

Kind regards, 

on behalf of

Lecturer Francesco Sessa 

Academic Editor

PLOS ONE